# Cost-effectiveness of repeat delayed imaging for spontaneous subarachnoid hemorrhage

**Wenru Shang**[1,2,3⊚]**, Huajie Jin**[3⊚]*****, Amisha Vastani**[4]**, Asfand Baig Mirza**[4]**, Benjamin Fisher**[4]**, Neeraj Kalra**[5]**, Ian Anderson**[5]**, Ahilan Kailaya-Vasan**[4]*****

**1** Evidence-Based Medicine Center, School of Basic Medical Sciences, Lanzhou University, Lanzhou, China, **2** WHO Collaborating Center for Guideline Implementation and Knowledge Translation, Lanzhou University, Lanzhou, China, **3** King's Health Economics (KHE), Institute of Psychiatry, Psychology & Neuroscience at King's College London, London, United Kingdom, **4** Department of Neurosurgery, King's College Hospital NHS Foundation Trust, London, United Kingdom, **5** Department of Neurosurgery, Leeds Centre for Neurosciences, Leeds General Infirmary, Leeds, United Kingdom

⊚ These authors contributed equally to this work.
* huajie.jin@kcl.ac.uk (HJ); a.kailaya-vasan@nhs.net (AKV)

## Abstract

### Background

In patients with intracranial aneurysm presenting with spontaneous subarachnoid hemorrhage (SAH), 15% of them could be missed by the initial diagnostic imaging. Repeat delayed imaging can help to identify previously undetected aneurysms, however, the cost-effectiveness of this strategy remains uncertain.

### Objective

The aim of this study is to assess the cost-effectiveness of repeat delayed imaging in patients with SAH who had a negative result during their initial imaging.

### Methods

A Markov model was developed to estimate the lifetime costs and quality-adjusted life-year (QALY) for patients who received or not received repeat delayed imaging. The analyses were conducted from a healthcare perspective, with costs reported in UK pounds and expressed in 2020 values. Extensive sensitivity analyses were performed to assess the robustness of the results.

### Results

The base case incremental cost-effectiveness ratio (ICER) of repeat delayed imaging is £9,314 per QALY compared to no-repeat delayed imaging. This ICER is below the National Institute for Health and Care Excellence (NICE) £20,000 per QALY willingness-to-pay threshold. At the NICE willingness-to-pay threshold of £20,000 per QALY, the probability that repeat delayed imaging is most cost-effective is 0.81. The results are sensitive to age, the utility of survived patients with a favorable outcome, the sensitivity of repeat delayed imaging, and the prevalence of aneurysm.

**Data Availability Statement:** All relevant data are within the paper and its Supporting Information files.

**Funding:** The authors received no specific funding for this work.

**Competing interests:** The authors have declared that no competing interests exist.

## Conclusions

This study showed that, in the UK, it is cost-effective to provide repeat delayed imaging using computed tomographic angiography (CTA) for patients with SAH who had a negative result in their initial imaging.

## Introduction

Subarachnoid hemorrhage (SAH), a severe type of hemorrhagic stroke, is characterized by substantial morbidity, mortality, and an imposing economic burden [1, 2]. It was estimated that the annual economic burden of SAH is £510 million in the UK [3–5]. The rupture of an intracranial aneurysm stands as the overwhelmingly predominant cause of spontaneous SAH [6, 7]. The mortality for patients with untreated aneurysmal SAH is up to 50% in the first month, mainly due to the consequences of re-rupture [8, 9]. Therefore, early identification and treatment of the underlying (causative) ruptured aneurysm is crucial. Digital subtraction angiography (DSA) and computed tomographic angiography (CTA) are commonly used diagnostic modalities for detecting and characterizing intracranial aneurysms [3, 10]. DSA is one of the most sensitive diagnostic tests and is the gold standard diagnostic test for the detection of intracranial aneurysms. However, DSA is expensive and invasive [11, 12]. It was reported that 0.5% of patients who received DSA experienced neurological complications [10]. Compared with DSA, CTA is less sensitive, but it is also less expensive and non-invasive [13]. The European Stroke Organization Guidelines recommend that CTA plus DSA should be used for the initial diagnosis of spontaneous SAH [3]. CTA plus DSA is also the most commonly used imaging strategy for patients with SAH in the UK. However, not all patients with an aneurysm can be identified by the initial diagnostic imaging [3, 14–16].

The European Stroke Organization Guideline recommends that if no aneurysm is identified during the initial imaging, then delayed imaging should be repeated using either CTA or DSA [3]. Some hospitals in the UK, such as the King's College Hospital, routinely provide a repeat delayed CTA to patients who had a negative result during their initial imaging. However, there is a lack of economic evidence about the cost-effectiveness of repeat delayed imaging in the UK or EU. To our knowledge, only two published economic evaluations assessed the cost-effectiveness of repeat delayed imaging, however, both studies were conducted in the US [17, 18]. In addition, neither of these two studies assessed patients with SAH who had a negative result during their initial imaging using both CTA and DSA.

To fill the evidence gap and to inform clinical decision-making in the UK, this study developed a Markov model to evaluate the cost-effectiveness of using a repeat delayed imaging strategy in SAH patients who initially had a negative result on CTA and DSA. This study was reported according to the Consolidated Health Economic Evaluation Reporting Standards 2022 (CHEERS 2022) Statement for reporting health economic evaluations [19].

## Materials and methods

A Markov model was developed in TreeAge Pro (Version 2020, TreeAge Software, Inc, Williamstown, Massachusetts) to estimate the cost-effectiveness of delayed repeating imaging versus no further imaging for a hypothetical cohort of 55-year-old patients with perimesencephalic SAH who had a negative result during their initial imaging. A lifetime horizon was adopted for the cost-effectiveness analysis, which means that the patients' health

outcomes and costs were calculated until their death. In line with the *European Stroke Organization Guidelines for the Management of Intracranial Aneurysms and Subarachnoid Hemorrhage* and the current UK practice, the initial imaging tests were assumed to be conducted using CTA followed by a DSA [3]. The repeat delayed imaging was assumed to be conducted using CTA, which is the clinical practice implemented at 'King's College Hospital, London. Since no individual patient information was utilized for this study, neither informed consent nor Institutional Review Board (IRB) approval was required.

This study applied the perspective of the NHS and Personal Social Services, as recommended by the National Institute for Health and Care Excellence (NICE) [20]. All costs were expressed in UK pounds (£) (2020 value). The primary measure of effectiveness in this study was quality-adjusted life-years (QALYs), a comprehensive metric that combines both the quality of life and survival aspects. Cost-effectiveness is quantified using the incremental cost-effectiveness ratio (ICER), which denotes the additional cost incurred per additional unit of QALY gained. The willingness-to-pay (WTP) threshold set by NICE is £20,000–30,000 per QALY, representing the pound value of a year in perfect health [20]. The intervention with an ICER lower than the predetermined WTP threshold is considered to be cost-effective.

## Model design

The model structure was shown in **Fig 1**. Patients in the "No repeat delayed imaging" arm (Fig 1A) would be discharged from the hospital following a negative result of their initial imaging using DSA and CTA. A proportion of these patients would remain free from re-bleeding until death. The cost and utility for those patients were assumed to be the same as the general population. Those patients who did not receive repeat delayed imaging and experience re-bleeding would require emergent treatment, and some patients may die before or during the emergent treatment. Patients who survived emergent treatment would enter a Markov process (Fig 1C), where 'patients' long-term prognosis was grouped into three health states based on the

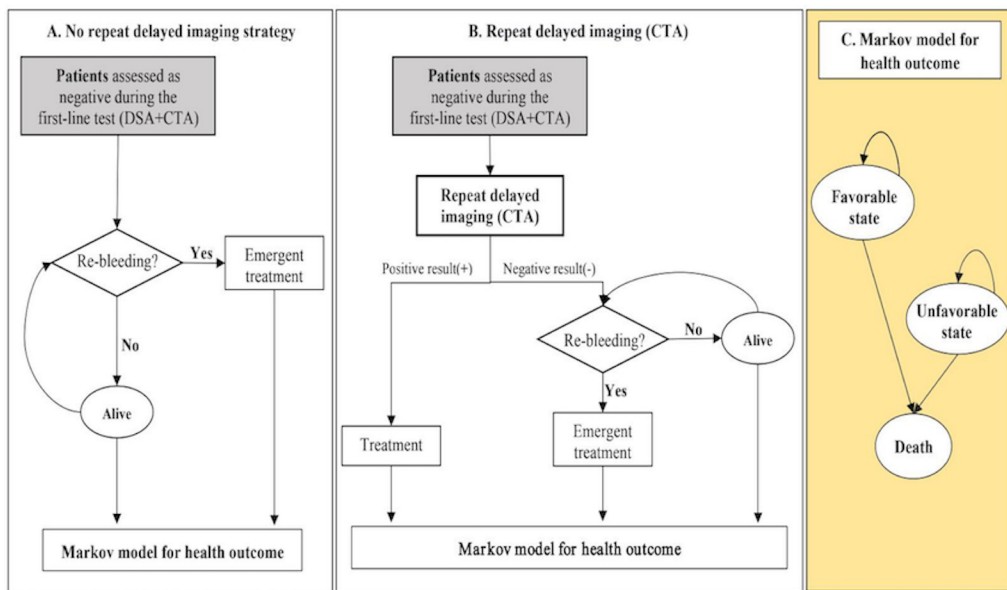

**Fig 1. The pathway of no repeat delayed imaging and repeat delayed imaging (CTA) for patients assessed as a negative result during the first-line test.** CTA, computed tomographic angiography; DSA, digital subtraction angiography. (An mRS score of 0 to 2 corresponded to a favorable state, mRS score of 3 to 5 corresponded to an unfavorable state, mRS score of 6 corresponded to death).

modified Rankin Scales (mRS): favorable (score 0 to 2), unfavorable (score 3 to 5), and death (score 6) [21]. The time period of the model is divided into cycles of time. After discussion with clinicians, a one-year cycle length was adopted for this study considering the growth rate of SAH, frequency of re-bleeding, the time interval adopted by the long-term follow-up studies, and computational burden of the model [22]. A one-year cycle length was also adopted by previous economic analysis assessing similar decision question [23]. At the end of each one-year cycle in the Markov process, individuals could either transition to a different health state or continue residing in their current state. Probabilities are assigned for either staying in the same state or transitioning to a different state within the model.

Patients in the "Repeat delayed imaging" arm (Fig 1B) would receive a repeat delayed imaging using CTA, seven days after their initial imaging. Patients were assumed to stay in the hospital during the 7-day interval between the initial and repeat delayed imaging. After receiving the repeat delayed imaging, those patients with a negative result would be discharged from the hospital without receiving any treatment. Those false negatives–patients with aneurysms that have not been identified by the initial or repeat delayed imaging–were assumed to be at continuous risk of re-bleeding and would receive emergent treatment if they did re-bleed. Patients found to harbor a vascular lesion during the repeat delayed CTA would receive a DSA to confirm the existence of an aneurysm before receiving non-emergent treatment, such as endovascular coiling and surgical clipping. The specificity of DSA, being the reference standard, was assumed to be 100% (i.e., zero false positive) [11]. Patients with an aneurysm who survived emergent or non-emergent treatment would enter the Markov process, as described above.

## Model parameters

Model input data were informed by published literature, supplemented by expert opinion (Table 1). The sensitivity and specificity of CTA were assumed to be 98.00% and 100%, based on a recent systematic review that assessed the accuracy of CTA in diagnosing cerebral aneurysms in patients with SAH [24]. The incidence of vascular abnormality in SAH patients with negative initial vascular imaging was assumed to be 17.3% based on a recent systematic review [16]. The annual probabilities of re-bleeding for SAH patients were obtained from a 10-year cohort study of 364 patients with cerebral aneurysms in England [25], shown in S1 Table. The transitional probabilities between different health states were taken from a 10-year follow-up of a UK cohort —the International Subarachnoid Aneurysm Trial—which reported the annual probability of favorable and unfavorable outcomes after aneurysm treatment [26], shown in S1 Table.

## Cost

Cost of delayed CTA imaging was obtained from the NHS reference cost 2018/2019 [35]. Cost of aneurysm treatment was obtained from published literature, which included the cost of Imaging, healthcare consumables, surgical equipment, blood products, and hospital bed occupancy [29]. Duration of hospital stay before the delayed imaging was assumed to be seven days. The annual treatment cost of nursing for unfavorable outcome states per year was obtained from the published literature [30]. No costs were assigned to patients in the "no re-bleeding and favorable state". All costs were discounted by 3.5% each year, in accordance with the National Institute for Health and Care Excellence (NICE) reference case [20].

## Utilities

The health state utility weights were obtained from published literature [30, 33]. Death was assigned a utility value of 0. The utility weight for the "no re-bleeding" state was assumed to be

**Table 1.  Model input parameters.**

| Description | Base-line value | Range | Distribution | Source |
|---|---|---|---|---|
| **Clinical parameters** | | | | |
| Age | 55 | 50 ∼ 80 | / | A |
| The prevalence of aneurysm (after initial negative imaging result) | 17.30% | 5.00% ∼ 35.00% | Beta (α = 5.26, β = 21.05) | [3, 15, 16] |
| Probability of functional outcome in re-bleeding SAH patients | | | | |
| favorable outcome | 0.0752 | —— | Dirichlet (0.0752, 0.3948, 0.4700) | [27] |
| unfavorable outcome | 0.3948 | —— | | [27] |
| Probability of functional outcome in no re-bleeding patients | | | | |
| favorable outcome | 0.51 | —— | Dirichlet (0.51, 0.34, 0.15) | [27] |
| unfavorable outcome | 0.34 | —— | | [27] |
| Probability of functional outcome in SAH patients with no aneurysm | | | | |
| favorable outcome | 0.8708 | 0.8030 ∼ 0.9160 | Beta (α = 1740.8400, β = 258.2900) | [28–30] |
| unfavorable outcome | 0.0792 | 0.0791 ∼ 0.0793 | Beta (α = 9.9500, β = 1246.2100) | [28–30] |
| The mortality of functional outcome | | | | |
| favorable outcome | 0.1067 | —— | Dirichlet (0.1067, 0.7778, 0.1155) | [31] |
| unfavorable outcome | 0.7778 | —— | | [31] |
| The accuracy for CTA | | | | |
| sensitivity | 98.00% | 20.00% ∼ 99.00% | Beta (α = 737.51, β = 15.05) | [24, 32] |
| specificity | 100.00% | 20.00% ∼ 100.00% | Beta (α = 247.50, β = 3.76) | [24, 32] |
| **Cost (£)** | | | | |
| CTA per test | 331.00 | ±25% | Gamma (α = 144.00, λ = 0.43) | A |
| DSA per test | 1,919.00 | ±25% | Gamma (α = 143.99, λ = 0.075) | A |
| Hospital stay during delay imaging (per day) [a] | 380.75 | ±25% | Gamma (α = 144.01, λ = 0.051) | [29] |
| Treatment for aneurysm [b] per patient (coiling) | 14,030.80 | ±25% | Gamma (α = 144.00, λ = 0.01) | [29] |
| Treatment for aneurysm [b] per patient (clipping) | 13,212.19 | ±25% | Gamma (α = 144.00, λ = 0.01) | [29] |
| Emergency treatment for aneurysm [b] per patient (coiling) | 19,983.87 | ±25% | Gamma (α = 144.00, λ = 0.007) | [29] |
| Emergency treatment for aneurysm [b] per patient (clipping) | 20,275.02 | ±25% | Gamma (α = 144.00, λ = 0.007) | [29] |
| Cost of nursing for unfavorable outcome states per year | 84,346.39 | ±25% | Gamma (α = 144.00, λ = 0.002) | [30] |
| The length of stay (LOS) for inpatient (day) | 7 | | | Assume |
| **Utility** | | | | |
| Utility of favorable outcome | 0.87 | 0.60 ∼ 1.00 | Beta (α = 20.82, β = 6.05) | [30, 33] |
| Utility of unfavorable outcome | 0.44 | 0.11 ∼ 0.71 | Beta (α = 3.82, β = 5.50) | [15, 33] |
| Utility of death | 0 | | | [34] |
| Utility of no re-bleeding states | 0.60 | 0.41–0.72 | Assumed to be the same as Stroke survivors [3] | |
| **Discounting** | | | | |
| Discount rate for QALYs | 0.035 | —— | —— | [20] |
| Discount rate for cost | 0.035 | —— | —— | [20] |

A. NIHR Industry Costing Template. Investigation and Intervention Tariff 2019/2020.

[a] It was assumed that before receiving repeated delayed imaging, patients were managed in vascular wards.

[b] Treatment for aneurysm included the cost of radiology, graft, blood, theatre, additional theatre, and hospital stay.

the same as that of stroke survivors with minor or moderate disability—defined as with some or significant restrictions in lifestyle [34]. All QALYs were discounted by 3.5% each year [20].

## Cost-effectiveness analysis

**Sensitivity analysis.**   Two types of sensitivity analysis were performed: deterministic one-way sensitivity analysis and probabilistic sensitivity analysis (PSA). Deterministic one-way

sensitivity analysis aimed to evaluate the influence of individual parameter variations on the model's outcomes. The range for sensitivity analysis was established by considering the upper and lower values of the variable as reported in the published literature. If these values were not reported, a range of ±25% of the base case value was applied for sensitivity analysis (Table 1). The Tornado diagram in Fig 2 showed the relative importance of key parameters on the ICER. The horizontal bars in the diagram represent the range of each parameter. For example, the range of the utility of a favorable outcome is 0.6–1. The width of each bar indicates the magnitude of its impact on the ICER–the longer the bar, the greater influence the parameter has on ICER [36]. As shown in Fig 2, the top four most influential variables to our model are age, the utility of a favorable outcome, sensitivity of CTA, and prevalence of aneurysm. PSA was conducted to assess the overall uncertainty in the cost-effectiveness of the model, taking into account the combined impact of all key parameters. Five thousand Monte Carlo simulation was conducted. The distributions used in the PSA are reported in Table 1. Cost-effectiveness acceptability curves (CEACs) were conducted to illustrate the likelihood that each intervention is cost effectiveness at various WTP threshold for an additional QALY.

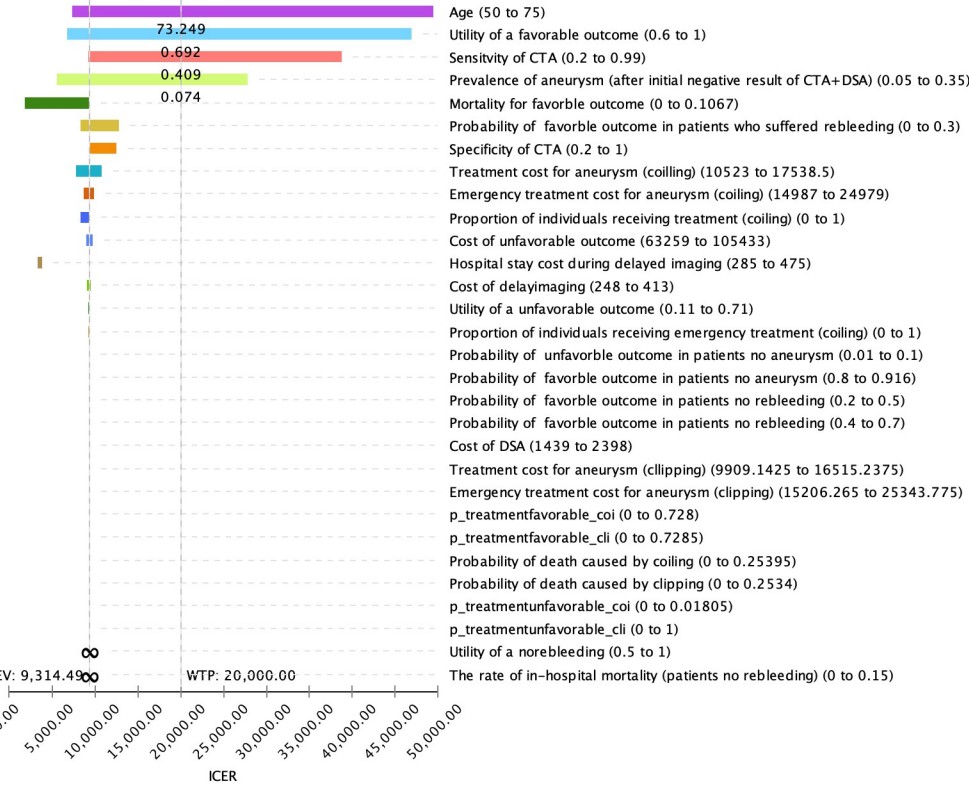

**Fig 2. A tornado diagram of 1-way sensitivity analyses to demonstrate the effects of varying parameters on the incremental cost-effectiveness ratio (ICER) for no second-line strategy vs delayed CTA imaging strategy.** The wider bars at the top have the greatest effect on the ICER, while variations in inputs at the bottom have small effects. The willingness-to-pay (WTP) line is at an ICER of £20,000. Numbers in parentheses after the variables are the parameter ranges. CTA, computed tomographic angiography; DSA, digital subtraction angiography; ICER, incremental cost-effectiveness ratio; QALY, quality-adjusted life-year.

## Results

### Base case scenario

In the base case scenario (Table 2), no repeat delayed imaging strategy was associated with an expected cost of £ 7,430 per patient and an expected effectiveness of 4.656 QALY per patient. Repeat delayed imaging strategy using CTA was found to be associated with an anticipated cost of £10,065 per patient and an expected effectiveness of 4.939 QALYs per patient. The ICER of the repeat delayed imaging strategy was £ 9,314 per QALY, which was below the NICE £ 20,000 ⁄per QALY WTP threshold.

### Sensitivity analysis

The tornado diagram in **Fig 2** illustrates the outcomes of one-way sensitivity analyses, specifically evaluating the effects of uncertainty associated with each input parameter on the ICER estimate of repeat delayed imaging using CTA in comparison to no repeat delayed imaging. The base case conclusion (repeat delayed imaging using CTA being the most cost-effective intervention) is sensitive to the following parameters:

- age of the patient increased to 73.25 years old (base case value: 55 years old)

- utility of favorable outcome decreased to 0.69 (base case value: 0.87)

- sensitivity of CTA decreased to 40.90% (base case value: 98.00%)

- prevalence of aneurysm in patients with SAH who had a negative result in their initial imaging decreased to 7.41% (base case value: 17.30%)

The results of the one-way sensitivity analysis for key parameters tested are reported in detail in the (S1 Fig).

The CEAC shown in **Fig 3** indicated that at the NICE WTP threshold of £20,000 per QALY, the probability of repeat delayed imaging being cost-effective is 81.05%.

## Discussion

Our results of cost-effectiveness analysis demonstrate that it is cost-effective to repeat delayed imaging using CTA for patients with SAH who had a negative result on their initial imaging, compared with no imaging. Although repeat delayed imaging was associated with an increased cost compared to no repeat delayed imaging, its additional cost can be justified by the additional QALY gains resulting from early identification and early treatment for patients with an aneurysm. The cost-effectiveness of the repeat delayed imaging strategy was most sensitive to age, the utility of favorable outcomes, the sensitivity of CTA, and the prevalence of aneurysms in patients with SAH who had a negative result in their initial imaging.

Before our study, the cost-effectiveness of repeat delayed imaging was only assessed by two US studies [17, 18]. Jethwa *et al.* assessed the cost-effectiveness of providing delayed DSA to

**Table 2. Result of cost-effectiveness (basic case scenario).**

| Strategy | Costs (£) | Effectiveness (QALYs) | Incremental Cost (£) | Incremental Effectiveness (QALYs) | Incremental C/E Ratio, ICER (£/QALY) |
|---|---|---|---|---|---|
| No repeated delayed imaging | 7,430 | 4.656 | - | - | - |
| Repeated delayed imaging using CTA[a] | 10,065 | 4.939 | 2635.29 | 0.283 | 9,314 |

[a] CTA, computed tomographic angiography.

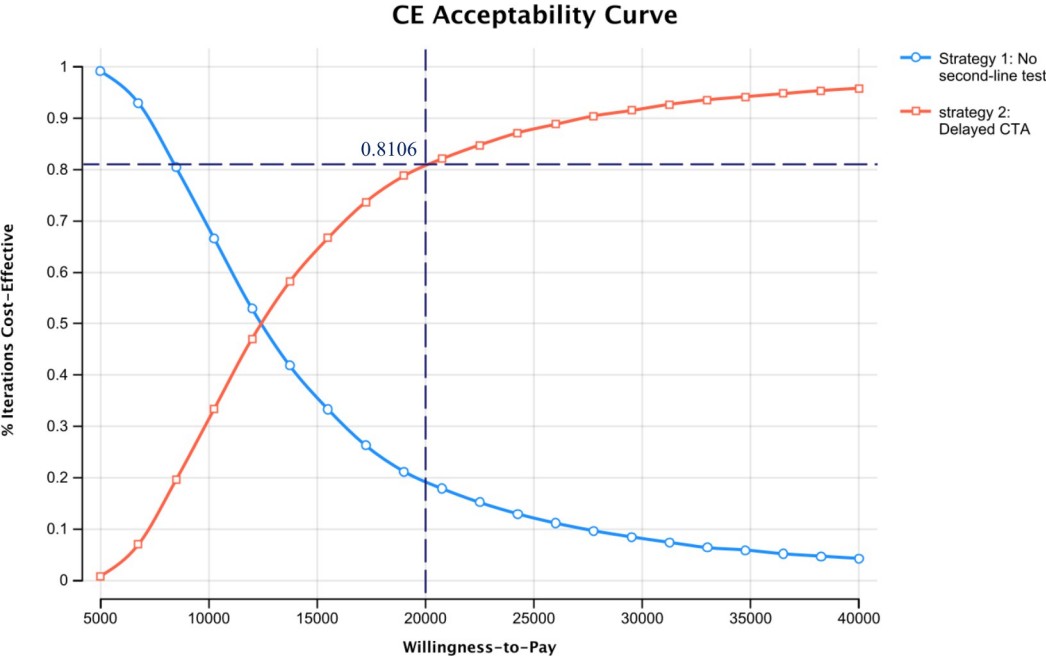

**Fig 3. Acceptability curve demonstrating the percentage of iterations that were cost-effective for each strategy over varied willingness-to-pay thresholds.** QALY, Quality Adjusted Life Year.

patients with SAH who were diagnosed as negative during their initial imaging using CTA. They found that delayed imaging (using DSA) is not cost-effective unless the negative predictive value of delayed imaging is above 93.73%. Kalra *et al*. compared twelve imaging strategies for both initial imaging and follow-up imaging in patients with perimesencephalic SAH. They found that conducting a CTA without follow-up imaging is the most cost-effective strategy. However, our model showed that compared with no repeat delayed imaging, repeat delayed imaging (using CTA) is cost-effective in patients with SAH who had a negative result during their initial imaging, as long as the sensitivity of CTA exceeded 40.90%. This might be because of three reasons. Firstly, Jethwa *et al*. used CTA as the first-line test and DSA as the second-line test, whilst our study used DSA+CTA as the first-line test and CTA as the second-line test. Secondly, both Jethwa *et al*. and Kalra *et al* adopted a short time horizon (i.e., only covered the time period from patients receiving the initial imaging to patients receiving a definitive diagnosis), whilst our study adopted a lifetime horizon. Therefore, the two US studies are likely to underestimate the QALY losses and future treatment costs for those patients with aneurysms who were missed by the initial imaging. If those lifetimes QALY losses and future treatment costs were considered, repeat delayed imaging would appear to be more cost-effective. Last but not least, due to the difference in financing mechanisms of healthcare system, payment structure and approaches to healthcare delivery, the price of healthcare services and procedures in the US is consistently higher than the price in the UK. According to the Organisation for Economic Co-operation and Development (OECD) Health Statistics 2022, the per capita health spending in the U.S. was more than two times higher than other developed countries, including the UK, Australia, France, Canada and New Zealand [37]. As a result, the cost-effectiveness results of studies conducted in the US might not be transferable to the UK setting.

## Implication for clinical practice and future research

Our findings support the provision of repeat delayed imaging using CTA to patients with SAH who had a negative result in their initial imaging compared to no imaging. Given CTAs can be done without the expertise of a neuro-interventionalist, unlike a DSA, this choice of imaging provides a quick and cost-effective method to exclude a potential aneurysm in this cohort of patients. We would hope this study generates further interest in the management of patients with SAH who had a negative result during their initial imaging and potentially a national guideline in the delayed radiological investigation of choice in identifying an intracranial aneurysm in these patients. Notably, the results of sensitivity analysis showed that the conclusion is most sensitive to patients' age, the utility of a favorable outcome and prevalence of aneurysm in patients with SAH who had a negative result in their initial imaging. These findings suggest that repeat delayed imaging might not be cost-effective for patients who are over 73 years old, have a quality of life lower than 0.69 after surviving SAH and achieved a favorable outcome, or when the prevalence of aneurysm is lower than 7.41%. Potential strategies to improve the quality of life for aneurysm survivors include developing a follow-up care plan with the patient, providing rehabilitation services, monitoring blood pressure and headaches, and promoting a healthy life-style that includes abstaining from smoking and engaging in regular exercise [38]. Further evidence on the utility of favorable outcomes for aneurysm survivors and local prevalence of aneurysm in patients with SAH who had a negative result in their initial imaging is needed to robustly demonstrate the cost-effectiveness of repeat delayed imaging.

## Strengths and limitations

Our study has three strengths. First, to our knowledge, this is the first lifetime cost-effectiveness analysis that directly compares the repeat delayed imaging with no imaging for patients with SAH who had a negative result in their initial imaging in the UK. Our findings have important implications for clinical practice and can improve the health outcomes of patients aneurysmal with SAH. Secondly, previous economic evaluations did not include the nursing cost of SAH patients, whilst our study considered the lifetime cost for patients with unfavorable outcomes. Thirdly, to assess the robustness of the conclusion under various input data scenarios, both deterministic and probabilistic sensitivity analyses were conducted.

There are a number of limitations of our study, so the results must be interpreted cautiously. First, there is a lack of nursing costs for patients who survived SAH with unfavorable outcomes in the UK. We had to use the nursing cost reported by a Dutch study (£ 84,346.39) for patients over age 70 as a proxy. Therefore, a wide range of nursing costs (£ 63,260 to £ 105,433) was tested using one-way sensitivity analyses, and the base case conclusion remained unchanged. Second, the potential complications of CTA, such as acute allergic reactions and contrast-induced nephropathy, were not included in our model due to a lack of long-term cost and health outcome data for patients experiencing those complications. However, existing evidence suggests that the risk of severe allergic reactions and contrast-induced nephropathy is extremely low or even negligible for patients who undergo CTA [39, 40]. Therefore, we do not anticipate that the omission of these complications would significantly alter the conclusions of our study. Third, given that the prevalence of aneurysms is likely to be low in those patients with SAH who had a negative result during their initial imaging, the accuracy of CTA might be affected. A wide range of sensitivity (20.00% ∼ 99.00%) and specificity (20.00% ∼ 100.00%) for CTA was tested using one-way sensitivity analyses. We found that the conclusion remains the same unless the sensitivity of CTA below 40.90%. Fourth, we assumed that the health utility for patients with SAH in the "no re-bleeding" health state was the same as the

utility for minor stroke survivors, due to a lack of utility data for the former patients. This assumption contradicts the use of repeat delayed imaging since the primary advantage of employing repeated tests is to identify patients with aneurysms earlier and mitigate the risk of re-bleeding. Five, as we only evaluated one repeat delayed imaging, the results were not explored on other delayed imaging techniques.

## Conclusions

Our analyses showed that it is cost-effective to repeat delayed imaging using CTA compared with no repeat delayed imaging for patients with SAH who had a negative result in their initial imaging in the UK. Further evidence on the utility of favorable outcomes for survivors of aneurysms and the local prevalence of aneurysms in patients with SAH who had a negative result in their initial imaging is needed to robustly demonstrate the cost-effectiveness of repeat delayed imaging.

## Supporting information

**S1 Fig. Results of one-way sensitivity analysis.**
(PDF)

**S1 Table. Model input parameters.**
(DOCX)

## Author Contributions

**Conceptualization:** Huajie Jin, Amisha Vastani, Asfand Baig Mirza, Benjamin Fisher, Ahilan Kailaya-Vasan.

**Data curation:** Wenru Shang, Amisha Vastani, Asfand Baig Mirza.

**Formal analysis:** Huajie Jin.

**Investigation:** Wenru Shang, Huajie Jin, Amisha Vastani, Asfand Baig Mirza, Neeraj Kalra, Ian Anderson, Ahilan Kailaya-Vasan.

**Methodology:** Wenru Shang, Huajie Jin, Benjamin Fisher.

**Project administration:** Wenru Shang.

**Software:** Wenru Shang.

**Supervision:** Huajie Jin, Ahilan Kailaya-Vasan.

**Validation:** Wenru Shang, Amisha Vastani, Asfand Baig Mirza, Neeraj Kalra, Ian Anderson, Ahilan Kailaya-Vasan.

**Writing – original draft:** Wenru Shang, Huajie Jin, Ahilan Kailaya-Vasan.

**Writing – review & editing:** Wenru Shang, Huajie Jin, Amisha Vastani, Asfand Baig Mirza, Benjamin Fisher, Neeraj Kalra, Ian Anderson, Ahilan Kailaya-Vasan.

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
