## [Decision Letter · Decision Letter 0]

10 May 2023

PONE-D-22-31987Cost effectiveness of repeat delayed imaging for spontaneous subarachnoid hemorrhagePLOS ONE

Dear Dr. Jin,

Thank you for submitting your manuscript to PLOS ONE. After careful consideration, we feel that it has merit but does not fully meet PLOS ONE’s publication criteria as it currently stands. Therefore, we invite you to submit a revised version of the manuscript that addresses the points raised during the review process.

We look forward to receiving your revised manuscript.

Kind regards,

Alfred Pokmeng See, M.D.

Academic Editor

PLOS ONE

- https://doi.org/10.1161/CIRCOUTCOMES.119.006406

In your revision ensure you cite all your sources (including your own works), and quote or rephrase any duplicated text outside the methods section. Further consideration is dependent on these concerns being addressed.

Additional Editor Comments:

Thank you Dr. Jin et al. The manuscript nicely models a cost-effectiveness of imaging. I would recommend modifications according to the attached reviewer comments.

Reviewers' comments:

Reviewer's Responses to Questions

**Comments to the Author**

1. Is the manuscript technically sound, and do the data support the conclusions?

Reviewer #1: Yes

Reviewer #2: Yes

2. Has the statistical analysis been performed appropriately and rigorously? 

Reviewer #1: Yes

Reviewer #2: I Don't Know

3. Have the authors made all data underlying the findings in their manuscript fully available?

Reviewer #1: Yes

Reviewer #2: Yes

4. Is the manuscript presented in an intelligible fashion and written in standard English?

Reviewer #1: Yes

Reviewer #2: Yes

5. Review Comments to the Author

Reviewer #1: This study evaluated the cost-effectiveness of repeat delayed imaging in patients with

SAH who had a negative result during their initial imaging from a healthcare perspective.

The manuscript was very well-written and logically constructed.

I totally agree to accept for publication, while some minor revisions are needed.

1.In abstract, the control group should be clarified in the description of results.

2.Some minor grammar errors need to be revised, for example, in line 110 should say ‘were’ instead of ‘was’.

3.The time horizon ‘lifetime’ may need to be clarified in methods section.

4.Complications for patients with SAH were not explicitly modeled in this study, the authors have illustrated this as a limitation, this is acceptable.

5.Line 130-136: Sensitivity Analysis: the range of sensitivity and specificity should be clarified.

Reviewer #2: The authors investigated the cost effectiveness of using repeat delayed imaging to diagnosis unruptured aneurysms that were missed after initial imaging. To do so, the authors developed a Markov model using available data relating to costs of imaging and treatment in the UK and the utility of certain outcomes. The authors presented their results as they related to the lifetime costs and QALY that patients were estimated to experience. This paper was well-written and easily digestible; however, the below points should be considered before the work is published.

1. Given that the model was specifically designed using assumptions for 55-year-old patients, it would be prudent for the authors to explicitly note this limitation in the ability to generalize their findings to a more diverse patient population.

2. Why did the authors choose to divide the time period of the study into one year cycles? A brief description of the reasoning and a relevant citation would help ground this decision.

3. The authors do not specify what aneurysm treatment method was used in the model (i.e. clipping versus coiling), which would be helpful in determining the overall applicability of their findings. Were both treatment methods accounted for or only one? If only one was factored into the analysis, this limitation should be noted.

4. The tornado diagram was difficult to understand; a more detailed explanation of how to interpret the diagram would be helpful in parsing the key points, as well as a higher-resolution image.

5. The authors mention two studies that were conducted in the US (Jethwa et al. and Kalra et al.) but do not comment on the fundamental differences in healthcare systems and overall healthcare costs between the two countries; the authors should consider briefly discussing these differences in the estimated costs used in the US studies versus their own UK study and how these differences, if any, may account for the difference in their findings.

6. Additionally, the authors note that one of the limitations of their study includes not explicitly modeling complications due to a lack of evidence. It is not clear to me what is meant by "lack of evidence". Are the authors referring to a lack of available data about the cost of complications from SAH in the UK? Given that Jethwa et al. includes complication-associated costs in their analysis, it may be prudent to explain in more detail how this limitation came about and the possible effect its exclusion may have on the results.

7. Finally, a more detailed explanation of how to interpret the utility and mortality of functional outcome would aid in understanding the results of the model and its sensitivity. For example, what factors may influence "the utility of survived patients with a favorable outcome"?

6. PLOS authors have the option to publish the peer review history of their article (what does this mean?). If published, this will include your full peer review and any attached files.

Reviewer #1: No

Reviewer #2: No

---

## [Author Response · Author response to Decision Letter 0]

15 Jun 2023

Please see response letter attached.

---

## [Editor Report · Decision Letter 1]

13 Jul 2023

Cost-effectiveness of Repeat Delayed Imaging for Spontaneous Subarachnoid Hemorrhage

PONE-D-22-31987R1

Dear Dr. Jin,

We’re pleased to inform you that your manuscript has been judged scientifically suitable for publication and will be formally accepted for publication once it meets all outstanding technical requirements.

Kind regards,

Alfred Pokmeng See, M.D.

Academic Editor

PLOS ONE

---

## [Editor Report · Acceptance letter]

17 Jul 2023

PONE-D-22-31987R1 

Cost-effectiveness of Repeat Delayed Imaging for Spontaneous Subarachnoid Hemorrhage 

Dear Dr. Jin:

I'm pleased to inform you that your manuscript has been deemed suitable for publication in PLOS ONE. Congratulations! Your manuscript is now with our production department. 

Kind regards, 

on behalf of

Dr. Alfred Pokmeng See 

Academic Editor

PLOS ONE